# Single-Center Experience in Detecting Influenza Virus, RSV and SARS-CoV-2 at the Emergency Department

**DOI:** 10.3390/v15020470

**Published:** 2023-02-08

**Authors:** Manfred Nairz, Theodora Todorovic, Clemens M. Gehrer, Philipp Grubwieser, Francesco Burkert, Martina Zimmermann, Kristina Trattnig, Werner Klotz, Igor Theurl, Rosa Bellmann-Weiler, Günter Weiss

**Affiliations:** Department of Internal Medicine II (Infectious Diseases, Immunology, Rheumatology, Pneumology), Medical University of Innsbruck, A-6020 Innsbruck, Austria

**Keywords:** COVID-19, emergency department, influenza virus, infection, multiplex RT-PCR, POCT, respiratory viruses, RSV, SARS-CoV-2

## Abstract

Reverse transcription polymerase chain reaction (RT-PCR) on respiratory tract swabs has become the gold standard for sensitive and specific detection of influenza virus, respiratory syncytial virus (RSV) and severe acute respiratory syndrome coronavirus 2 (SARS-CoV-2). In this retrospective analysis, we report on the successive implementation and routine use of multiplex RT-PCR testing for patients admitted to the Internal Medicine Emergency Department (ED) at a tertiary care center in Western Austria, one of the hotspots in the early coronavirus disease 2019 (COVID-19) pandemic in Europe. Our description focuses on the use of the Cepheid^®^ Xpert^®^ Xpress closed RT-PCR system in point-of-care testing (POCT). Our indications for RT-PCR testing changed during the observation period: From the cold season 2016/2017 until the cold season 2019/2020, we used RT-PCR to diagnose influenza or RSV infection in patients with fever and/or respiratory symptoms. Starting in March 2020, we used the RT-PCR for SARS-CoV-2 and a multiplex version for the combined detection of all these three respiratory viruses to also screen subjects who did not present with symptoms of infection but needed in-hospital medical treatment for other reasons. Expectedly, the switch to a more liberal RT-PCR test strategy resulted in a substantial increase in the number of tests. Nevertheless, we observed an immediate decline in influenza virus and RSV detections in early 2020 that coincided with public SARS-CoV-2 containment measures. In contrast, the extensive use of the combined RT-PCR test enabled us to monitor the re-emergence of influenza and RSV detections, including asymptomatic cases, at the end of 2022 when COVID-19 containment measures were no longer in place. Our analysis of PCR results for respiratory viruses from a real-life setting at an ED provides valuable information on the epidemiology of those infections over several years, their contribution to morbidity and need for hospital admission, the risk for nosocomial introduction of such infection into hospitals from asymptomatic carriers, and guidance as to how general precautions and prophylactic strategies affect the dynamics of those infections.

## 1. Introduction

Respiratory viruses can cause a spectrum of manifestations including asymptomatic infection, common cold, tracheobronchitis, pneumonia and severe respiratory distress syndrome (SARS) [1,2,3,4]. They can circulate as endemics in communities, which have no or incomplete immunity, and can be the cause of epidemics or, in the worst-case scenario, pandemics [5,6,7,8].

In the northern hemisphere, influenza A and B virus as well as respiratory syncytial virus (RSV) cause seasonal bursts in the fall and winter time, as do four seasonal human coronaviruses (HCoV), i.e., HCoV-229, HCoV-HKU1, HCoV-NL63 and HCoV-OC43 [9,10]. In addition, pandemic viruses have emerged, such as the pandemic 2009 H1N1v influenza A virus, SARS-coronavirus-1 (SARS-CoV-1) and SARS-CoV-2 [11,12,13,14,15]. SARS-CoV-2 arose in late 2019 and caused a pandemic that was officially declared on 11 March 2020 and remains ongoing as of December 2022 [16,17,18,19]. Therefore, SARS-CoV-2 continues to exert many adverse effects on society, including economic, medical, political and social ones [20].

Different strategies are in use to test for SARS-CoV-2, including diagnostic, screening and public health surveillance testing [21,22]. In the management of medical emergencies, both diagnostic and screening tests are established options. Diagnostic testing aims at identifying subjects who have current symptoms and signs of respiratory tract infection to formulate the correct diagnosis, detect co-infections and initiate specific antiviral therapy and cohort isolation. Screening tests intend to find asymptomatic individuals who may carry SARS-CoV-2 and may have been recently exposed to the virus. The latter approach is helpful in guiding cohort isolation in the hospital, where the number of physical beds is limited, and in minimizing the risk of in-hospital transmission and nosocomial outbreaks.

In a diagnostic laboratory environment, biomedical experts perform RT-PCR under standardized and quality-controlled conditions [23]. Several steps, including the serial review of each RT-PCR result by technicians and physicians, result in substantial turn-around times and costs, though. Therefore, closed RT-PCR systems, which are easy to use, have been developed for point-of-care testing (POCT) [24]. One of these is the Cepheid^®^ Xpert^®^ Xpress system, which can be used by a trained operator such as a healthcare professional. Its deployment enables the simple and fast detection of clinically relevant pathogens such as influenza virus, RSV and SARS-CoV-2 using specific test cartridges. Further, at the end of the RT-PCR reaction, the software algorithm classifies the result as viral RNA detected, undetected or undetermined, simplifying the review process. Rapid antigen tests are also widely used for POCT. For example, they are employed by nurses in the emergency department (ED). However, they are typically less sensitive and less specific than RT-PCR [25,26,27,28]. This can cause false-negative or false-positive test results, both of which are especially problematic in the management of medical emergencies [27,29].

The University Hospital Innsbruck is the only public hospital in Innsbruck, the regional capital of Tyrol with about 130,000 inhabitants. At the same time, it serves as a tertiary center that covers more than 1.1 million inhabitants of Tyrol and Vorarlberg, the two most western regions in Austria. For these reasons, the spectrum of clinical conditions treated at the Internal Medicine ED is very broad and ranges from common respiratory infections to severe respiratory failure requiring extracorporeal membrane oxygenation (ECMO). Our hospital has therefore become one of the centers in Central Europe for the treatment of patients with severe influenza, RSV and COVID-19 requiring intensive care management and ECMO. To formulate the correct diagnosis in the Internal Medicine ED, we replaced rapid antigen testing by RT-PCR using the Cepheid^®^ Xpert^®^ Xpress system as POCT with 24/7 availability. The analysis presented here covers the time interval from 1 June 2016 to 23 January 2023. We report on the changes in the detections of influenza virus, RSV and SARS-CoV-2 before and during the COVID-19 pandemic, when public measures were initiated to limit its spread, and an endemic situation was approaching.

## 2. Materials and Methods

### 2.1. Specimen Collection

All respiratory specimens were collected either in BD^®^ or Copan^®^ universal viral transport medium (UTM) by trained healthcare professionals and analyzed or stored within 10 min after collection. Following written instructions, healthcare professionals took either a nasopharyngeal or oropharyngeal swab, depending on the lead symptom and anatomy of the test subject.

### 2.2. Test Strategy

Generally, the decision to perform RT-PCR for influenza virus, RSV or SARS-CoV-2 was at the discretion of the attending physician in the Internal Medicine ED.

Before March 2020, the indication for RT-PCR testing for influenza virus and RSV was mainly based on symptoms. Typically, patients with fever and/or respiratory symptoms were tested during the cold season or when they had a recent travel history to a country with epidemic influenza. During peaks of seasonal influenza epidemics, some attending physicians may have made the diagnosis of influenza based on clinical and epidemiologic criteria [30,31] or a positive rapid antigen test. After March 2020, patients without fever and respiratory symptoms were also tested for SARS-CoV-2 in the Internal Medicine ED when they were likely to require in-hospital treatment. Hospital staff was advised not to use the RT-PCR POCT for self-testing to spare resources.

### 2.3. Immunologic Assays

As a rapid antigen test, we used the Quidel^®^ Sofia^®^ Influenza A + B fluorescent immunoassay (FIA) between CW 05/2016 and CW 08/2019 [32].

### 2.4. RT-PCR

Cepheid^®^ system: The limit of detection (LoD) of the Xpert^®^ Xpress SARS-CoV-2 and Xpert^®^ Xpress SARS-CoV-2/Flu/RSV assays for SARS-CoV-2 ranges between 8 and 64 copies/mL according to the literature [33,34]. The Xpert^®^ Xpress Flu/RSV assay has a LoD of 0.013–0.75 influenza A TCID_50_/mL (for median tissue culture infectious dose), 0.19–0.40 influenza B TCID_50_/mL, 0.87–1.10 RSV A TCID_50_/mL and 0.79–2.30 RSV B TCID_50_/mL according to the instructions for use (IFU). The Xpert^®^ Xpress SARS-CoV-2 assay has a LoD of 0.005 and 0.02 PFU/mL (for plaque-forming units) for the nucleocapsid-2 (N2) and envelope (E) gene, respectively, according to the IFU. The Xpert^®^ Xpress SARS-CoV-2/Flu/RSV assay has a LoD of 131 SARS-CoV-2 copies/mL, 0.004–0.087 influenza A TCID_50_/mL, 0.04 influenza B TCID_50_/mL, 0.43 RSV A TCID_50_/mL and 0.22 RSV B TCID_50_/mL according to the IFU.

Altona^®^ system: Before the Cepheid^®^ Xpert^®^ Xpress SARS-CoV-2 kit became available, we ran the RealStar^®^ SARS-CoV-2 RT-PCR Kit 1.0 for research use only (RUO) between CW 11/2020 and CW 15/2020. This kit detects the spike (S) gene of SARS-CoV-2 and the E gene of the lineage B-betacoronarivurs with a LoD of approximately 625–1200 copies/mL sample for each of the genes [35,36]. We stored respiratory specimens collected in UTM within 10 min for up to 16 h at 4 °C. Samples were run batchwise 2 to 3 times per day in the ID laboratory, and no RT-PCR was available for SARS-CoV-2 between CW 11/2020 and CW 15/2020 from 9 p.m. to 7 a.m.

### 2.5. Data Analysis

Data were collected in the Laboratory Information System (LIS) of the Infectious Diseases (ID) laboratory as part of the routine diagnostic work flow. For the analysis presented herein, data were extracted from the LIS, sorted in Microsoft^®^ Excel and further analyzed in GraphPad^®^ Prism. A season was defined as the time interval from 1 June to 31 May of the following year. The cold season 2022 was analyzed from 1 June to 30 November 2022. Each CW was analyzed from 8:00 a.m. Monday to 7:59 a.m. on the Monday of the following week. CW 48/2022 was the last week considered in this analysis and was defined as the time interval from 8:00 a.m. on 28 November to 7:59 a.m. on 5 December. For Figure 2, Figure 5 and Figure 7, as well as for the analysis of co-infections, test results were considered until 7:59 a.m. on 23 January. The total number of respiratory specimens analyzed between 1 June 2016 and 23 January 2023 was 39,614. For the comparison of prepandemic and pandemic/endemic number of tests and virus RNA detections, we used the Mann–Whitney U-test calculated in GraphPad^®^ Prism version 9.

## 3. Results

During the observation period from 1 June 2016 to 5 December 2022, we gradually replaced rapid antigen tests in the Internal Medicine ED and Infectious Diseases (ID) laboratory with RT-PCR (Table 1).

We introduced RT-PCR for influenza, RSV and/or SARS-CoV-2 at the point of care (POC) as soon as specific test cartridges became available. In the cold seasons 2016/2017 and 2017/2018, documented proportions of 19.3% and 13.3% of all tests performed to detect influenza virus were rapid antigen tests (Figure 1a).

Starting in the cold season 2018/2019, we used only the Cepheid^®^ Xpert^®^ Xpress system and test cartridges for influenza virus (FLU) and RSV. After a short time interval from calendar week (CW) 11 to CW 15/2020, when we could only run RT-PCR for SARS-CoV-2 batchwise using the Altona^®^ RealStar^®^ RT-PCR system, we started to use test cartridges for SARS-CoV-2 in CW 16/2020. The emergence of SARS-CoV-2 in early 2020 caused a substantial increase in the absolute number of all tests performed for individuals in the ED (Figure 1b).

This stepwise introduction of POC RT-PCR systems with SARS-CoV-2 coverage resulted in a steady increase in the number of tests performed (Figure 2a). The number of tests performed reflected the seasonal circulatory pattern of influenza virus and RSV until CW 10/2020. After the emergence of SARS-CoV-2 in Western Austria, we performed RT-PCR for SARS-CoV-2 year-round. We started in CW 11/2020 with a SARS-CoV-2 single test and later also used the combined Cepheid^®^ SARS-CoV-2/influenza/RSV test. The effort to detect and isolate asymptomatic SARS-CoV-2-carrying individuals, who required in-hospital treatment, resulted in a dramatic increase in the number of RT-PCR tests performed and respiratory viruses detected (Figure 2b).

When we compared the number of RT-PCR tests for respiratory viruses in Internal Medicine ED patients, we saw a significant increase in the number of tests per CW before and during the COVID-19 pandemic (Figure 3a). In parallel, the number of weekly virus RNA detections increased as well, which was also affected by the severity of the different SARS-CoV-2 surges (Figure 3b).

The number of RT-PCR tests increased for all three respiratory viruses, with RT-PCR for SARS-CoV-2 performed most frequently (Figure 4a). SARS-CoV-2 also dominated viral RNA detections (Figure 4b). In contrast, the numbers of influenza virus and RSV detections decreased from the cold season 2020/2021 onward. In this context, it is important to note that the analysis presented in Figure 4 was performed at the end of CW 48/2022, at a time when the number of infections with RSV and influenza A was rising, and thus only covers 6 months of the cold season 2022/2023 (indicated as ‘2022’ in Figure 4a,b).

When we analyzed the virus detections in each CW, we saw that the seasonal pattern of influenza virus and RSV was perfectly represented in the number of detections at the Internal Medicine ED until CW 10/2020 (Figure 5). Furthermore, influenza virus and RSV were not detected in subjects undergoing RT-PCR testing at the Internal Medicine ED in the cold season 2020/2021.

After the installation of containment measures (Table 2) beginning on 10 March 2020 in response to the early spread of SARS-CoV-2 in Western Austria, we saw a dramatic decrease in influenza virus detections on 14 March (Figure 6a).

Intriguingly, no influenza virus or RSV was detected in the Internal Medicine ED for more than 14 months between 14 March 2020 and 31 May 2021. In contrast, the effects of the containment measures on SARS-CoV-2 detections were smaller and occurred later (Figure 6b). On 25 March, 7 days after the quarantine of local communities was ordered, the detection of SARS-CoV-2 in subjects tested at the Internal Medicine ED started to decline as well. Throughout April 2020, no more than two SARS-CoV-2 detections per day were documented in the Internal Medicine ED (Figure 6c).

Thereafter, we detected influenza A virus in 26 patients in CW 22 and 23/2021, which is an unusual cumulation in the warm season that had not been observed in previous years. Notably, none of these subjects had detectable SARS-CoV-2 in the respiratory sample. However, 7/26 were tested for suspected COVID-19 (26.9%), and 19/26 were tested as screening for SARS-CoV-2 (73.1%). Interestingly, the first detections were recorded 7 days after the Pentecost holidays, which are preferred travel holidays in Austria, including for international travel. A further and more substantial reemergence of influenza virus detections did not occur before CW 07/2022.

When comparing the peak of influenza virus detections during epidemics in the cold seasons from 2016/2017 to 2021/2022, we saw that these peaks occurred in CW 02/2017, CW 08/2018, CW 07/2019, CW 52/2019 and CW 14/2022. Therefore, after the emergence of SARS-CoV-2, the pattern of respiratory virus detections of Internal Medicine ED test subjects changed. Specifically, there was no evidence of an influenza epidemic in the cold season 2020/2021, yet there was a small peak in CW22/2021 attributable to a total of 26 influenza A virus detections in the warm season. Moreover, the peak of influenza virus detections in the cold season 2021/2022 was smaller and occurred later than in the seasons before the emergence of SARS-CoV-2 and containment measures. Importantly, there were >100 weekly influenza virus detections between CW 49/2022 and CW 52/2022, indicating a massive epidemic of seasonal influenza in the cold season 2022/2023. Similarly, we recorded >20 weekly RSV detections between CW 50/2022 and CW 01/2023 as well as in CW 03/2023 in subjects admitted to the Internal Medicine ED. Following the final analysis of our data set after CW 03/2023 and the revision of this manuscript, influenza A detections continued to decrease, whereas RSV detections remained around 20 per CW (Figure 5).

The increase in the number of RT-PCR performed and the more liberal RT-PCR test strategy including specific regulations of the health authorities in response to the SARS-CoV-2 pandemic resulted in alterations of the detection rates for influenza virus and RSV (Figure 7).

The peak detection rates per CW for influenza virus between cold seasons 2016/2017 and 2019/2020 ranged between 39.3 and 50.0% and—after the emergence of COVID-19—generally dropped to rates of 0.0% in the cold season 2020/2021 and 5.6% in the cold season 2021/2022 (Figure 7a), with a burst in the detection rate to 73.9% in the warm season 2021.

The peak detection rates per CW for RSV showed much more variability and were 41.7% in the cold season 2018/2019 and 6.6% in the cold season 2019/2020. Thereafter, detection rates per CW declined to 0% in the cold season 2020/2021 and 1.0% in the cold season 2021/2022 (Figure 7b).

SARS-CoV-2 detections rates per CW had numerous peaks of 7.3% in CW 11/2020, 35.2% in CW 43/2020, 10.9% in CW 12/2021, 21.4% in CW 47/2021, 39.1% in CW 11/2022, 28.0% in CW 29/2022 and 33.2% in CW 41/2022 (Figure 7c), which is consistent with the pandemic spread of SARS-CoV-2 and the peak incidences of the various epidemiologic surges.

## 4. Discussion

At the beginning of the SARS-CoV-2 pandemic, we witnessed a rapid increase in the number of RT-PCR tests performed at the Internal Medicine ED from CW11 to CW33/2020 in comparison to previous years. We assume that in this first phase of RT-PCR testing for SARS-CoV-2, this was largely attributable to the high number of adult patients seeking medical attention in the ED because of symptoms of respiratory tract infection including COVID-19. At the same time, the number of RT-PCR tests available was rather limited, and mainly symptomatic individuals were tested for several reasons. On the one hand, SARS-CoV-2 RT-PCR systems were at different stages of development, approval, production and global distribution. On the other hand, there was a trend in many countries for extensive testing of symptomatic and asymptomatic individuals, including close contacts of COVID-19 patients. The ID laboratory is hospital-based. Therefore, we chose to prioritize RT-PCR testing of patients with respiratory symptoms. This strategy saved material in order to have enough supplies for diagnostic testing of subjects who were moderately to critically ill and thus required in-hospital or ICU treatment. Screening testing on a large scale, on the other hand, could have generated additional shortages in the supplies of swab sets and PCR reagents, as observed on the global market [39,40,41,42]. In CW34 through 43/2020, the number of weekly RT-PCR tests for respiratory viruses performed and the Internal Medicine ED remained below 100 (mean 52; range 23–77). This may be attributable to the effects of containment measures and the low activity of SARS-CoV-2 in the population during the warm season 2020, thus marking the second phase of the pandemic observation period of our study.

The Cepheid^®^ Xpert^®^ Xpress SARS-CoV-2 test cartridge, for which we have had the platform in use at the Internal Medicine ED for several years, only received emergency use authorization in the US on 30 March 2020 and did not become available in the ID laboratory before 17 April 2020. A few weeks later, however, in CW 44/2020, marking the beginning of the third wave of the pandemic, a more liberal test strategy in comparison to previous seasons was used due to regulations introduced by health authorities, but also in an attempt to avoid the spread of infections introduced by pre-symptomatic patients with COVID-19. In fact, from CW 44/2020 to CW 48/2022, a mean of 276 weekly tests (range 113–402) were performed in the Internal Medicine ED, which may be due to adaptations in the test strategy, the availability of a sufficient, yet rationed, number of test cartridges for our POCT system and to supply-induced demand. During the pandemic, the RT-PCR POCT strategy at the Internal Medicine ED fundamentally changed. Prior to the emergence of SARS-CoV-2, RT-PCR testing was largely symptom-based. At the peak incidences of the seasonal influenza epidemics, ED doctors may have even chosen to make the diagnosis of influenza based on clinical and epidemiologic criteria [30,31]. This strategy allows for the correct clinical diagnosis during epidemics with 70–80% accuracy [43]. Since the emergence of SARS-CoV-2, however, the clinical differential diagnosis between influenza and COVID-19 has become more difficult, because few symptoms and hematologic or biochemical laboratory results are specific for either disease [44,45,46,47,48]. Therefore, most if not all adults with symptoms of respiratory tract infection but also with non-respiratory symptoms putatively related to COVID-19 have undergone RT-PCR testing at the Internal Medicine ED since the beginning of the COVID-19 pandemic, especially if they required in-hospital care or antiviral therapy. Moreover, attending physicians were advised to also screen subjects without infectious or respiratory symptoms of SARS-CoV-2 by RT-PCR before admission to the hospital. Notably, after approval of the combined SARS-CoV-2/influenza/RSV test, this was more readily available to the ID laboratory and the Internal Medicine ED than the SARS-CoV-2 single test cartridge. Therefore, we tested a large number of subjects with and without respiratory symptoms for all three pathogens starting in CW 51/2020. In symptomatic patients, the spread of SARS-CoV-2 and the subsequent introduction of therapeutics may have resulted in an additional increase in the number of RT-PCR for respiratory viruses for differential diagnosis between the entities. Therefore, various reasons may have contributed to the increase in RT-PCR tests for respiratory viruses performed in the Internal Medicine ED when comparing the time before and during the COVID-19 pandemic.

Our data set provides much information about the circulation of influenza and RSV in our region in and after the cold season 2020/2021. Specifically, between CW 51/2020 and CW 21/2021, a total of 2382 combined SARS-CoV-2/FLU/RSV tests resulted in zero detections of influenza virus and RSV. Our data are thus in accordance with a recent study from Greece, which observed no detection of influenza virus in 2021 [49], and a study from Spain that reported no influenza or RSV detections in children between CW 14/2020 and CW 19/2021 [50]. Similarly, a study from Madagascar reported no detection of influenza virus and reduced RSV detections in the cold season 2020/2021 [51].

Between CW 22/2021 and CW 21/2022, a total of 4,990 combined SARS-CoV-2/influenza/RSV tests resulted in 95 detections of influenza virus and 5 detections of RSV. When comparing the 95 detections of influenza virus to previous seasons from 2016/2017 to 2019/2020, the relative reduction in detections amounted to 15.2–59.1%, despite more extensive testing. When comparing the five detections of RSV to the previous seasons 2018/2019 and 2019/2020, this relative reduction in detections amounted to 70.6–89.8%. This reduction in RSV and influenza infections may be explained by the efficacy of contact precautions and hygiene education of people along with travel restrictions in the seasons 2020/21 and 2021/22 (Figure 5). Accordingly, the lifting of these precautions and possibly also a weaning immunity [52,53,54] resulted in a drastic increase in the detection of those infections in late 2022. This was accompanied by an increasing number of co-infections between the cold season 2016/2017 and the ongoing season 2022/2023. Specifically, we detected 1 co-infection with influenza A virus and RSV in the season 2018/2019, 11 with SARS-CoV-2 and influenza A virus in the season 2021/2022, as well as 20 with SARS-CoV-2 and influenza A virus, 2 with SARS-CoV-2 and RSV and 7 with influenza A and RSV in the ongoing season 2022/2023. This corresponds to rates of co-infections of 0.7% for SARS-CoV-2, of 2.4% for influenza A and of 3.7% for RSV with either of the two other viruses. Therefore, the rate of co-infections of individuals with detectable SARS-CoV-2 was lower at our center than the rates of co-infections of 7.2% reported in patients <18 years of age in a pediatric ED in Lithuania [55] and of 9.9% observed in two tertiary centers in Bulgaria [56]. We assume that the low rate of co-infections detected in our ED is related to the fact that our study cohort only included patients ≥18 years of age, and that in Austria, some containment measures were still in place at the end of 2022 (Figure 5).

Our descriptive report has both strengths and weaknesses. A major strength is its large size of 39,614 respiratory samples and the extensive use of the combined SARS-CoV-2/FLU/RSV RT-PCR, which enabled us to obtain a comprehensive picture of the activities of all three viruses in our region between CW 51/2020 and CW 03/2023. We saw a rapid decline in influenza virus detection after the introduction of COVID-19 containment measures, which reflected the nationwide situation well [57]. Notably, we recorded 26 cases of influenza A in the warm season 2021, all of which were detected within 8 days in CW 22 and 23/2021. No such pattern was documented in the European Surveillance System (TESSy), as only 34 influenza detections were reported across the whole WHO European Region from CW 20/2021 to CW 28/2021 [58]. In the same 8-day time window, only five infections with SARS-CoV-2 were detected. It is interesting to note that the first of the influenza A virus detections occurred within 7 days after the Pentecost holidays and therefore may be attributable to international travels. Still, there were no further influenza virus detections after these 8 days until CW 04/2022, suggesting that the influenza virus did not further circulate in our region in the warm season 2021.

In the cold season 2022/2023, the first influenza virus detections in the Internal Medicine ED were documented in CW 44 and CW 45/2022, and dramatic rises in detections occurred between CW 47/2022 and CW 01/2023. In fact, the weekly influenza A virus detections recorded in the Internal Medicine ED in this time interval were the highest number ever documented in the laboratory information system (LIS) of the ID laboratory. Around the same time, the National Influenza Network Austria reported a slight increase in nationwide influenza detections from CW 44 to 45/2022 and roughly weekly increases of 50–100% from CW 45 to CW 50/2022. These comparisons suggest that the respiratory virus detections in the Internal Medicine ED have a similar sensitivity for the epidemiologic situation across Austria and better reflect the regional circulation of respiratory viruses in the western part of the country than nationwide recording. Further, as a hospital-based medical laboratory located in the regional capital, the ID laboratory has a favorable geographic and strategic position at the University Hospital Innsbruck as well as an adequate test frequency. Therefore, we propose that the ID laboratory can serve as sentinel laboratory for the sensitive monitoring of the circulation of respiratory viruses in the regional population.

Our study has several limitations, too. First, we extracted the data from the LIS and analyzed it retrospectively. Therefore, clinical information on our test subjects and the indication for testing were not systematically recorded and could not be analyzed in all subjects tested. Second, we only considered RT-PCR results from the Internal Medicine ED and ID laboratory. On the one hand, not all patients diagnosed in the ID laboratory were admitted to the hospital, but they are represented in the data analysis. On the other hand, patients diagnosed outside the ID laboratory and then transferred to a ward outside the Internal Medicine Department are not represented in our analysis, unless an additional sample taken upon transfer or another follow-up sample yielded detectable viral RNA.

Third, we only considered patients ≥18 years tested in the Internal Medicine ED, whereas children and adolescents <18 years were tested in the Pediatric ED and are thus not represented in our analysis. Nevertheless, our data may be important for the management of our hospital and for health authorities to plan resources required for the in-patient care of adult subjects.

Fourth, we did not determine the SARS-CoV-2 variants in the respiratory specimens tested in the ID laboratory. However, we have deduced from nationwide surveillance programs that the WT strain was the predominant variant until February 2021 [59,60]. In March 2021, the Alpha variant (B.1.1.7) became the most prevalent circulating virus until June 2021. From July 2021 to January 2022, the Delta variant (B.1.617.2) was the most frequently encountered variant. Starting in February 2022, Omicron (B.1.1.529) and its subvariants became the dominant variants in Austria.

Surveillance for respiratory tract infections can be performed at different levels, including regional, national and international levels. Typically, academic institutions or health authorities have established and maintain such surveillance systems, including the Diagnostic Influenza Network Austria [61], the Austrian RSV Network [62], the National Reference Centre for Influenza Viruses at the Robert Koch Institute in Germany and the European Influenza Surveillance Network [63]. Strictly speaking, the ID laboratory is not part of such a surveillance system, although it does report notifiable infectious diseases such as COVID-19, influenza H5N1, legionellosis and tuberculosis to the Austrian health authorities via an electronic reporting system.

From both an epidemiologic and a clinical standpoint, one of the most intriguing aspects of our data is that the containment measures introduced by health authorities resulted in a sharp decline in the detection of influenza and RSV to zero in less than one week. In contrast, the decline in the detection of SARS-CoV-2 was much flatter, required additional and more stringent containment measures and took several weeks. The reasons for these differences are hard to grasp, but may be attributable to several epidemiologic and virologic factors, such as specific differences between SARS-CoV-2 and influenza virus and/or RSV in the effective reproductive number [64,65,66,67,68,69,70], the role of overdispersion for its transmission [71,72,73,74,75,76,77], the mean viral load [73,78], the duration of virus shedding [79,80,81] and the minimal infective dose [75,82,83,84,85,86]. In addition to these and other pathogen-related factors, host- and population-related factors such as immunity to infection may be important. For example, in the cold season 2019/2020, there was a substantial degree of immunity against the epidemic viruses of influenza and RSV in the general population through previous exposures or vaccinations [72,87,88,89,90,91]. In contrast, only a small percentage of the population had antibodies against SARS-CoV-2 in March 2020 [6], and previous infections against seasonal human coronaviruses provide little cross-protection against SARS-CoV-2 [92,93]. The early COVID-19 pandemic has thus hit a fully susceptible general population, and the first SARS-CoV-2 vaccines did not become available in Tyrol before January 2021. Therefore, there was little immunity against SARS-CoV-2 in March 2020, as evident from the 1.1–3.1% seroprevalence of SARS-CoV-2 in Austria between April and June 2020 [94,95,96,97,98], possibly delaying the effects of containment measures alone on SARS-CoV-2 transmission.

Similar scenarios have been observed in influenza pandemics. In 2009, for instance, the pandemic H1N1, a virus of swine origin, spread throughout the world, which was partly attributable to the lack of cross-protective humoral immunity from seasonal influenza epidemics [99].

The number of SARS-CoV-2 detections by RT-PCR declined after March 2021. This may partly be attributable to the effects of the ongoing vaccination program. Between March and September 2021, the coverage of SARS-CoV-2 vaccination in subjects >16 years increased from approximately 4 to 58% in the region of Tyrol [100]. This was accompanied by a reduction in the detections of SARS-CoV-2, whereas the numbers of RT-PCR run in the ID laboratory did not substantially change. This suggests that a high coverage of the adult Tyrolean population with SARS-CoV-2 vaccination but not molecular testing was associated with a reduction in infections. However, other interventions and moderate temperatures may well have contributed to this reduction.

Despite numerous uncertainties, SARS-CoV-2, specifically the Omicron (B.1.1.529) variant, influenza virus and RSV, may have similarities in terms of symptoms, immune responses, laboratory parameters, the mean viral loads of affected individuals [101,102,103,104,105,106,107,108,109,110], the rate of asymptomatic infections and the protective effect of face masks [111,112], although the duration of viral shedding may be longest in SARS-CoV-2 infections [78,79,80].

In summary, our real-life data from an Internal Medicine ED in Western Austria demonstrate that influenza and RSV detections by RT-PCR were substantially reduced in the cold seasons 2020/2021 and 2021/2022, when COVID-19 containment measures were in place. In the same observation period, SARS-CoV-2 detections in our center showed several peaks that reflected well the nationwide situation of COVID-19 surges. After the lifting of most COVID-19 containment measures, we witnessed a reemergence of influenza virus and RSV with unprecedently high numbers of detections, whereas SARS-CoV-2 detections remained at moderate levels. These results highlight the importance of RT-PCR testing for SARS-CoV-2, influenza virus and RSV for the diagnosis and cohort isolation of adults presenting to an ED with or without symptoms of respiratory tract infection and for the observation of the regional epidemiologic situation. Further studies will be required to see if this information necessary to manage patients at the POC is also relevant and sufficient to adjust healthcare resources.

## Figures and Tables

**Figure 1 viruses-15-00470-f001:**
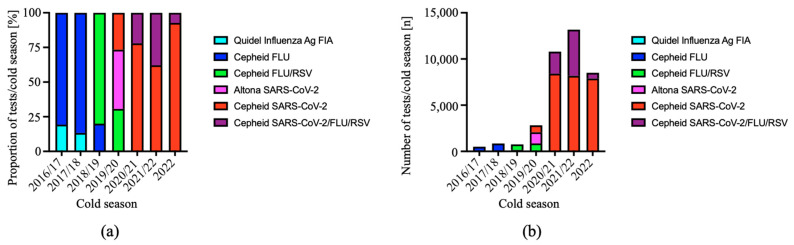
Use of POC test system for respiratory viruses in the Internal Medicine ED. We introduced RT-PCR testing for respiratory viruses step by step in the Internal Medicine ED. Seasons 2016/2017 to 2021/2022 were analyzed from 1 June to 31 May of the following year (12 months). The observation period in 2022 only included the interval from 1 June to 30 November (6 months). The relative proportion (**a**) and absolute number of tests (**b**) performed with each of the test systems is depicted.

**Figure 2 viruses-15-00470-f002:**
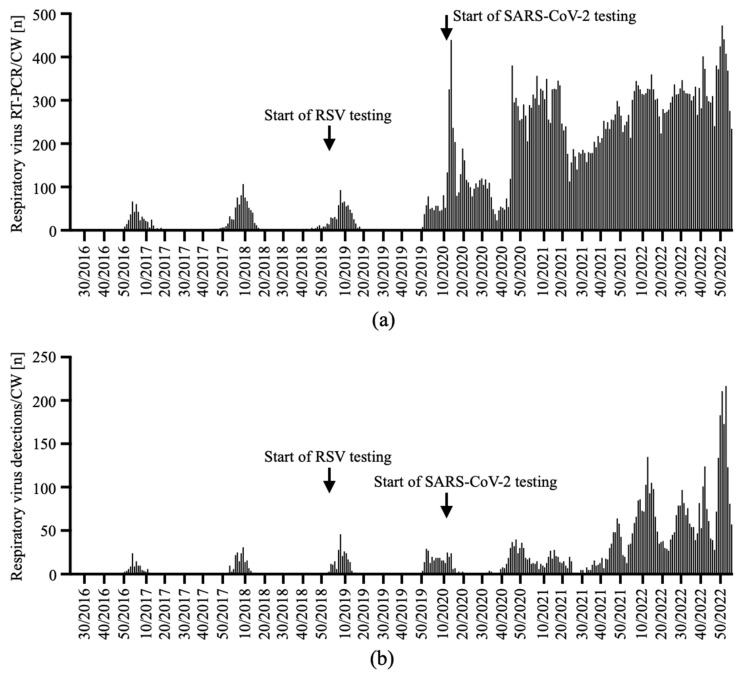
The total number of tests and detections for respiratory viruses. We analyzed the number of tests for influenza virus, RSV and SARS-CoV-2 in total. The number of tests (**a**) and detections (**b**) is depicted for each CW from CW 22/2016 to CW 03/2023. Each CW was analyzed from 8:00 a.m. Monday to 7:59 a.m. on the Monday of the following week. Arrows denote the start of RSV and SARS-CoV-2 testing.

**Figure 3 viruses-15-00470-f003:**
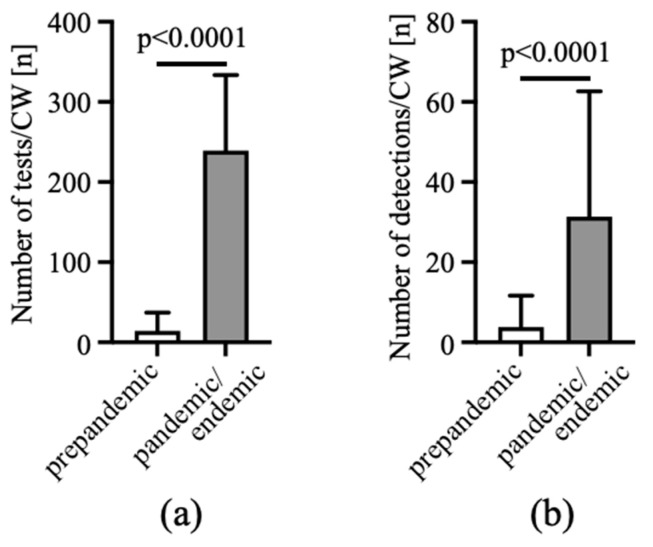
The SARS-CoV-2 pandemic caused a massive increase in RT-PCR tests for respiratory viruses and virus detections. The number of tests (**a**) run for subjects admitted to the Internal Medicine ED and the number of virus detections (**b**) were analyzed per CW. Results for the prepandemic time interval from CW 23/2016 to CW 10/2020 and for the pandemic and endemic time from CW 11/2020 to CW 48/2022 were compared. n = 197 prepandemic CW, n = 143 pandemic/endemic CW. Mann–Whitney U test.

**Figure 4 viruses-15-00470-f004:**
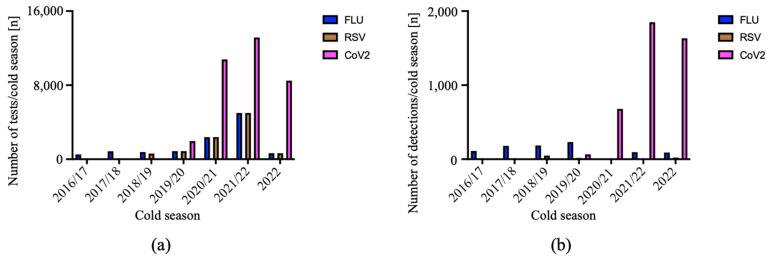
Number of tests for respiratory viruses and of virus detections in the Internal Medicine ED. We analyzed the number of tests (**a**) and of virus detections (**b**) for each of the respiratory viruses, influenza A and B virus (FLU in blue), RSV (RSV in brown) and SARS-CoV-2 (CoV2 in magenta) from the cold season 2016/2017 to 2022, the latter considered until 30 November 2022.

**Figure 5 viruses-15-00470-f005:**
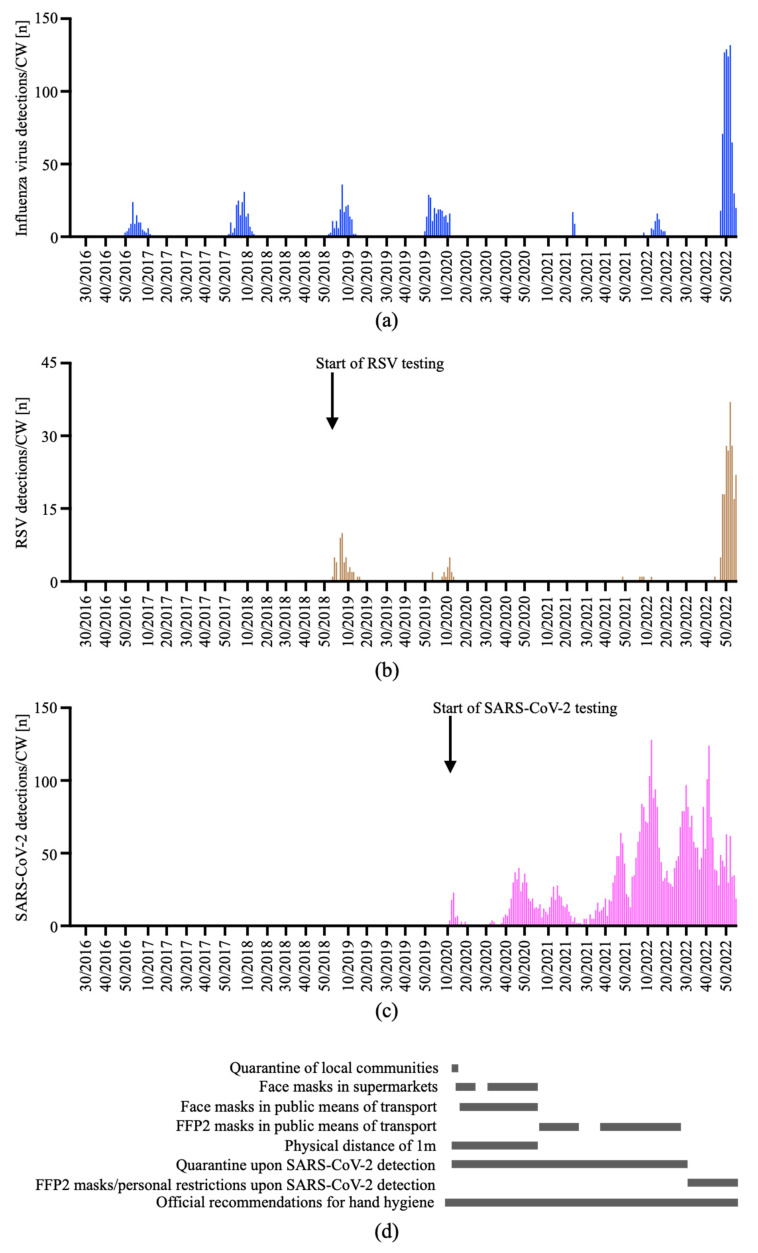
Number of virus detections. The number of influenza A or B virus (**a**), RSV (**b**) and SARS-CoV-2 (**c**) detections in individuals admitted to the Internal Medicine ED were analyzed for each CW from CW 22/2016 to CW 03/2023. The start of RSV and SARS-CoV-2 testing (arrows in (**a**–**c**)) as well as selected official recommendations and regulations for SARS-CoV-2 containment (bars in (**d**)) are indicated.

**Figure 6 viruses-15-00470-f006:**
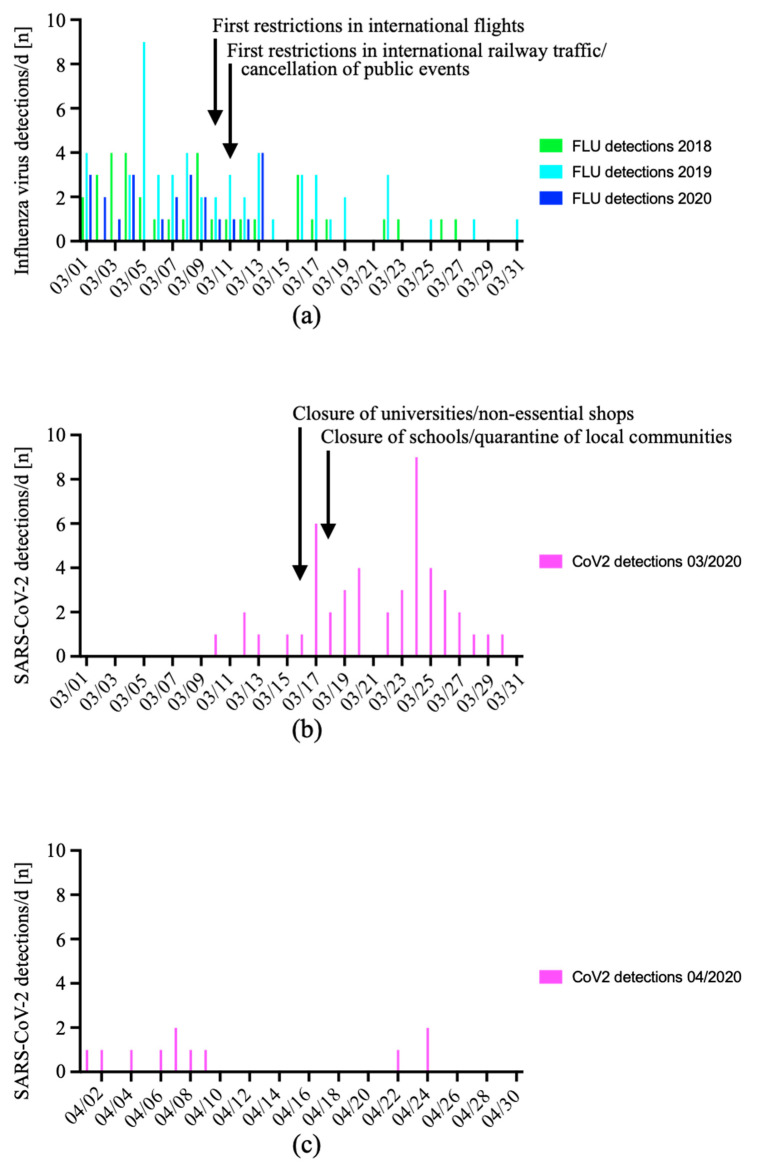
Effect of public COVID-19 containment measures on the daily detections of influenza virus. The detections of influenza A or B virus (**a**) in the Internal Medicine ED were analyzed in March 2020 (blue). Numbers of March 2018 (green) and March 2019 (turquoise) served as comparison. The detections of SARS-CoV-2 in March 2020 (**b**) and April 2020 (**c**) are depicted, too. The start of selected containment measures is indicated.

**Figure 7 viruses-15-00470-f007:**
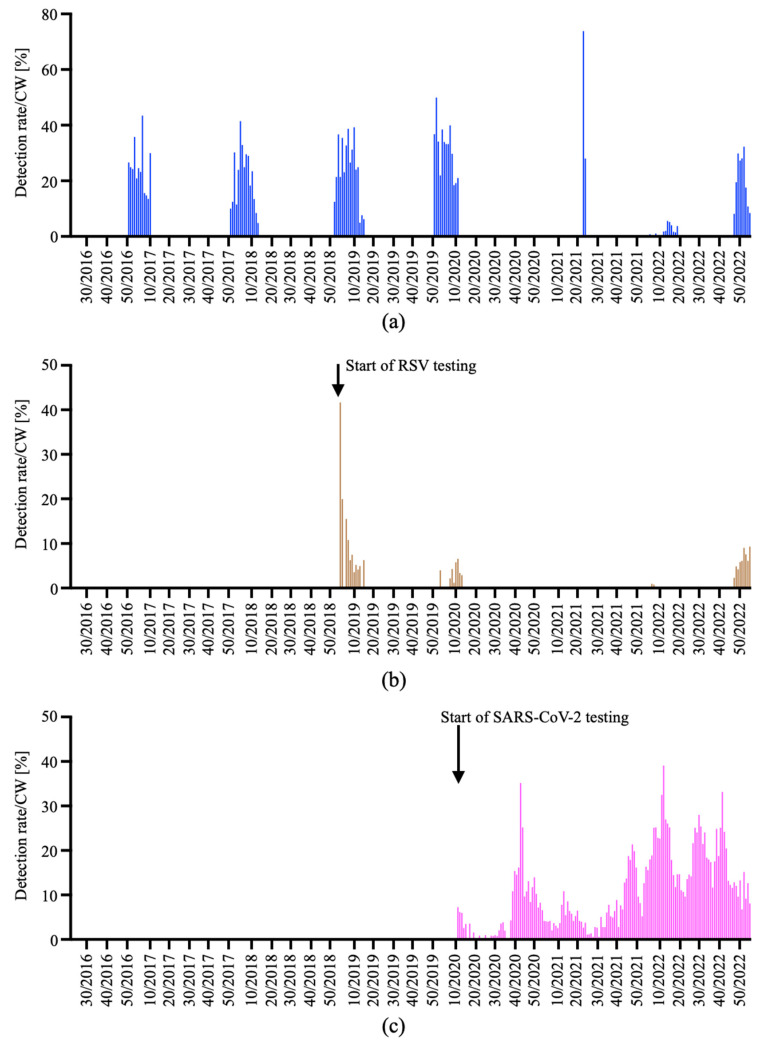
Detections rates per CW. Detection rates of influenza A or B virus (**a**), RSV (**b**) and SARS-CoV-2 (**c**) were only analyzed in CW, in which ≥10 tests were ordered. CW with <10 tests ordered were considered unrepresentative and were not depicted.

**Table 1 viruses-15-00470-t001:** Test systems used for respiratory viruses in Internal Medicine ED patients.

Name	Method	Format	Virus	Start (CW)	Stop (CW)	Application
Quidel^®^ Sofia^®^ Influenza	Ag test	FIA	Influenza A + B	05/2016	08/2019	Seasonal
Cepheid^®^ FLU	RT-PCR	Cartridge	Influenza A + B	30/2016	02/2019	Seasonal
Cepheid^®^ FLU/RSV	RT-PCR	Cartridge	Influenza A + B + RSV	01/2019	Ongoing	Seasonal
Altona^®^ SARS-CoV-2	RT-PCR	96-well plate	SARS-CoV-2	11/2020	15/2020	Temporary
Cepheid^®^ SARS-CoV-2	RT-PCR	Cartridge	SARS-CoV-2	16/2020	Ongoing	Perennial
Cepheid^®^ SARS-CoV-2 /FLU/RSV	RT-PCR	Cartridge	Influenza A + B + RSV + SARS-CoV-2	51/2020	Ongoing	Perennial

Table 1 lists the sequence in the use of test systems for respiratory viruses at the Internal Medicine ED. Abbreviations: Ag (antigen), CW (calendar week), FIA (fluorescent immunoassay), FLU (influenza virus), RSV (respiratory syncytial virus), RT-PCR (reverse transcription polymerase chain reaction), SARS-CoV-2 (severe acute respiratory syndrome coronavirus 2).

**Table 2 viruses-15-00470-t002:** Selected public containment measures in Austria.

Containment Measure	Start (Date)	Stop (Date)
First restrictions of international flights	03/10/2020	03/24/2020
First restriction of international railway traffic	03/11/2020	06/02/2020
Cancellation of public events	03/11/2020	05/10/2020
Closure of universities	03/16/2020	05/01/2022
Closure of non-essential shops	03/16/2020	05/01/2020
Restrictions of movement	03/16/2020	05/01/2020
Closure of gastronomy	03/17/2020	05/15/2020
Closure of schools	03/18/2020	05/17/2020
Quarantine of all local communities in Tyrol	03/18/2020	04/07/2020
Face masks in supermarkets and pharmacies	03/30/2020	06/08/2020
Face masks in public means of transport	04/14/2020	01/25/2021
Face masks in non-essential shops	05/06/2020	06/08/2020
FFP2 masks in public indoor places	01/25/2021	07/01/2021

Table 2 lists a selection of the first public containment measures in Austria [37,38]. Abbreviations: FFP2 (filtering facepiece class 2 according to the European standard EN 149).

## Data Availability

Data can be obtained from the corresponding authors on reasonable request.

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
