# Peer review of "Single-Center Experience in Detecting Influenza Virus, RSV and SARS-CoV-2 at the Emergency Department"

_viruses, 2023, doi:10.3390/v15020470_

Round 1

Reviewer 1 Report

Interesting and sound work which clearly merits publication. 

Major:

- The authors argue that we are still in a pandemic situation. However, there is some evidence that we have entered the endemic situation regarding SARS-CoV-2. This might require adjustment of Figure legends. 

- The authors might want to include a bar in Fig 5 which indicates when which hygiene measure had to be carried out

- The authors might want to include information on the predominant SARS CoV2 variant present at the respective periods of time

- The authors might want include some data on bacterial respiratory infections (if available)

- Fig. 1 left panel y axis: instead of numbers percent of total is given. Please adjust

- The authors need to discuss other very recent papers on this topic. 

Minor:

- All abbreviations should be detailled (TCID, LoD, CW).

Author Response

First of all, we would like to thank all reviewers for the overall very positive evaluation of our manuscript. Their insightful comments and recommendations gave us the opportunity to further improve our manuscript during the revision.

Reviewer 1

Interesting and sound work which clearly merits publication. 

Major:

- The authors argue that we are still in a pandemic situation. However, there is some evidence that we have entered the endemic situation regarding SARS-CoV-2. This might require adjustment of Figure legends.

We have adjusted figures and figure legends accordingly and have added a sentence at the end of the ‘Introduction’ and ‘Results’ sections.

- The authors might want to include a bar in Fig 5 which indicates when which hygiene measure had to be carried out

We have modified Figure 5 and are now depicting the duration of selected recommendations and regulations on hygiene measures for SARS-CoV-2 containment in the new panel d.

- The authors might want to include information on the predominant SARS CoV2 variant present at the respective periods of time

The SARS-CoV-2 variants were not determined in the respiratory specimens included in our study. However, from nationwide surveillance programs we deduce that the WT strain was the predominant strain until February 2021. In March 2021, the Alpha variant (B.1.1.7) became the most prevalent variant until June 2021. From July 2021 to January 2022, the Delta variant (B.1.617.2) was the most frequently encountered variant. Starting in February 2022, Omicron (B.1.1.529) and its subvariants became the dominant variants in Austria.

We now include this information as new paragraph in the ‘Discussion’ section.

- The authors might want include some data on bacterial respiratory infections (if available)

Unfortunately, we cannot provide data on bacterial respiratory infections because these are tested in another Department by culture methods.

- Fig. 1 left panel y axis: instead of numbers percent of total is given. Please adjust

Corrected to ‘Proportion’.

- The authors need to discuss other very recent papers on this topic. 

We have added several recent papers to the ‘Discussion’ section:

Frontiers in Medicine. 2023 Jan 9;9:1025147. doi: 10.3389/fmed.2022.1025147.

Viruses. 2022 Dec 20;15(1):12.  doi: 10.3390/v15010012.

Children. 2022 Sep 24;9(10):1464.  doi: 10.3390/children9101464.

Children. 2023 Jan 7;10(1):126.  doi: 10.3390/children10010126.

Frontiers in Public Health. 2022 Sep 2;10:959319.  doi: 10.3389/fpubh.2022.959319.

Minor:

- All abbreviations should be detailled (TCID, LoD, CW).

We have completed the details on the abbreviations used.

Reviewer 2 Report

This study reported the changes in the detections of influenza virus, RSV and SARS-CoV-2 before and after the COVID-19 pandemic at a tertiary center in Western Austria. 

There are several concerns that the authors should address in current manuscript:

(1).   What is already known on this topic, and what does your manuscript add?

(2). SARS-CoV-2 will circulate with other respiratory viruses, increasing the probability of co-infections. How many patients with SARS-CoV-2 were co-infected with other pathogens?

(3).    Line 89, Specimen collection: In this retrospective study, how many individuals were sampled and information was collected should be described in detailed.

(4).    Line 158: Ag (antigen); CW (calendar week); please omit “…”

(5).    Line 235: Intriguingly, no influenza or RSV (RS virus or RSV?) was detected…the manuscript should have been read and edited extensively.

(6).    Concerning the length of the Discussion: I strongly recommend that the authors substantially shorten the Discussion (especially the limitations of this study). This would improve the overall readability of your article.

(7).    Line 548: It is high recommend that the authors used a paragraph to summarize their findings.

Author Response

First of all, we would like to thank all reviewers for the overall very positive evaluation of our manuscript. Their insightful comments and recommendations gave us the opportunity to further improve our manuscript during the revision.

Reviewer 2

This study reported the changes in the detections of influenza virus, RSV and SARS-CoV-2 before and after the COVID-19 pandemic at a tertiary center in Western Austria. 

There are several concerns that the authors should address in current manuscript:

(1).   What is already known on this topic, and what does your manuscript add?

To address this point, we have added several recent studies to the discussion. We now also emphasize that we provide real-life data from a total of 39,614 respiratory samples collected at an Internal Medicine Emergency Department.

Our data set is thus very comprehensive, because all patients likely to be admitted were tested, and its updated version extending to calendar week 03/2023 now also includes the putative peaks of the virus endemics in the cold season 2022/2023. Little information on the cold season 2022/2023 has been published before.

(2). SARS-CoV-2 will circulate with other respiratory viruses, increasing the probability of co-infections. How many patients with SARS-CoV-2 were co-infected with other pathogens?

To address this point, we checked out dataset for co-infections with SARS-CoV-2, influenza A, influenza B and RSV. The data are reported in a new paragraph in the ‘Discussion’ section. We also included the most recent data from calendar week 49/2022 to calendar week 03/2023, a time when influenza and RSV epidemics peaked in the cold season 2022/2023.

(3).    Line 89, Specimen collection: In this retrospective study, how many individuals were sampled and information was collected should be described in detailed.

A total of 39,614 respiratory samples were tested. The data were collected in the laboratory information system (LIS) in the Infectious Disease laboratory as part of the routine diagnostic work-up. Data were exported, put in order in Excel and statistically analyzed in Prism. We have clarified these processes in the ‘Materials and Methods’ section.

(4).    Line 158: Ag (antigen); CW (calendar week); please omit “…”

We have changed the paragraph accordingly.

(5).    Line 235: Intriguingly, no influenza or RSV (RS virus or RSV?) was detected…the manuscript should have been read and edited extensively.

We have changed the wording to ‘influenza virus and RSV’ throughout.

(6).    Concerning the length of the Discussion: I strongly recommend that the authors substantially shorten the Discussion (especially the limitations of this study). This would improve the overall readability of your article.

We have shortened the discussion and added a paragraph on co-infections to make it both more focused and more relevant.

(7).    Line 548: It is high recommend that the authors used a paragraph to summarize their findings.

We now conclude our ‘Discussion’ section with a short paragraph to summarize our findings.

Reviewer 3 Report

The authors described their single-center experience using the Cepheid Xpert Xpress closed RT-PCR system to detect influenza virus, RSV, and SARS-CoV-2 in their emergency department. The manuscript is overall well written. I only have one minor suggestion:

1. The discussion seems a bit lengthy and unfocused. Can the authors please condense and shorten their discussion section to make it more focused?

Author Response

First of all, we would like to thank all reviewers for the overall very positive evaluation of our manuscript. Their insightful comments and recommendations gave us the opportunity to further improve our manuscript during the revision.

Reviewer 3

The authors described their single-center experience using the Cepheid Xpert Xpress closed RT-PCR system to detect influenza virus, RSV, and SARS-CoV-2 in their emergency department. The manuscript is overall well written. I only have one minor suggestion:

  1. The discussion seems a bit lengthy and unfocused. Can the authors please condense and shorten their discussion section to make it more focused?

We have shortened the discussion and added a paragraph on co-infections to make it both more focused and more relevant.

Round 2

Reviewer 2 Report

The authors have addressed all my comments. I have no further questions

The authors have addressed all my comments. I have no further questions